# Advertising between Archetype and Brand Personality

**Clemens Bechter [1,*], Giorgio Farinelli [2], Rolf-Dieter Daniel [3] and Michael Frey [4]**

[1]  Thammasat Business School, Thammasat University, Tha Prachan, 10200 Bangkok, Thailand
[2]  EuroMBA, Tongersestraat 49, 6211 LM Maastricht, the Netherlands; Giorgio.Farinelli@euromba.org
[3]  European Association for Business and Commerce, 1 Empire Tower, Sathorn Road, 10120 Bangkok, Thailand; rolf-dieter.daniel@eabc-thailand.eu
[4]  Faculty of Humanities, University of Freiburg, Friedrichstr 39, 79098 Freiburg, Germany; Michael@bagan.net
[*]  Correspondence: clemensbechter@tbs.tu.ac.th; Tel.: +66-2-623-5742

**Abstract:** The aim of the paper is the alignment of C.G. Jung's (1954) archetypes and Aaker's (1997) brand personality framework in the context of advertising. C.G. Jung's theories had a tremendous impact on psychology. David Aaker and his daughter Jennifer are seen by many as the branding gurus. Despite the fact that both frameworks refer to persons/personalities there is no publication linking the two frameworks. Our research tried to fill this gap by developing a joint framework combining Jung's and Aaker's attributes and apply it by analyzing two distinctively different TV commercials from Asian hotel chains. A total of 102 Executive MBA students had to watch both TV commercials and then conduct an Archetype (C.G. Jung) Indicator test and rate Brand Personality (Aaker) traits of the two commercials. Results show that there is common ground. This has implications for advertisers who may want to specify an archetype and related personality attributes for their promotional campaigns. Game changers in the hospitality sector may want to be seen as Outlaw whereas established hotel chains may position themselves as Lover with personality attributes such as welcoming, charming, and embraced.

**Keywords:** archetypes; promotion; branding; brand personality traits; positioning of hotels

## 1. Introduction

Established brands face the challenge of maintaining consumer's interest; one solution is the built-up of a specific brand personality [1]. Brand Personality consists of a number of human characteristics associated with the brand; it is a personification of the brand [2]. Matzler *et al.* [3] used a sample of 662 car enthusiasts and proved that personality traits extraversion and agreeableness predict identification with the brand community, which in turn, and along with product attachment, is related to trust and brand loyalty. Product attachment itself was a function of person-brand congruity, the perceived fit between the person and the brand. Allen and Olson [4] consider that brand personality is the set of meanings that best describe fundamental brand characteristics. These meanings are constructed by consumers based on behaviors seen in brands when they are personified or based on their attributes, in our case the two hotel chains. Brand personality and human personality share similarities: both are durable and might help predict the actions of buyers [5].

The concept of brand personality has been criticized on a conceptual level (what exactly is a brand personality?) as well as on a methodological and substantive level (how should it be defined and how does it differ from brand/user imagery?) [6]. Allen and Olson [4] addressed these three issues by viewing brand personality from a narrative perspective which helps understanding the processes

by which consumers form personality impressions e.g., brand characters. Research by Padgett and Allen [7] suggests that narratives are highly effective in communicating service experiences.

Consumers tend to express their own personality either actual or idealistic with the products they buy [8]. It is essential to understand what kind of personality traits are associated with a brand and what kind of self-projection occurs when consumers buy a brand. Mulvey and Medina [9] found that a considerable portion of the meaning of an ad is derived from the characters (which can be human or animated) in the ad.

Brand Personality can play an important role in the consumer choice linked to self-expression in the sense of 'this is me' [10] as well as a strategic brand positioning tool [11]. The brand personality concept can be applied to anything from a product or service to a whole country [12] or tourism destination [13].

Aaker [2] linked the five dimensions of human personality [14] to Brand Personality Traits, see Table 1.

Table 1. Human Dimensions and Brand Traits (adapted from [2]).

| Human Dimension | Brand Personality Traits |
|---|---|
| Sincerity | Down to Earth<br>Honest<br>Genuine<br>Cheerful |
| Excitement | Daring/Adventure<br>Spirited<br>Imaginative<br>Up-to-date |
| Competence | Reliable<br>Responsible<br>Dependable<br>Efficient |
| Sophistication | Glamorous/Upper Class<br>Pretentious<br>Charming |
| Ruggedness | Romantic<br>Tough<br>Strong<br>Outdoorsy<br>Rugged |

The five dimensions correspond with the Big 5 of personality structure [14]. As such it is not a novel approach. It has been criticized for confusing user profiles (e.g., upper class) with brand characteristics. It has also been criticized for its weak discriminatory power [15,16].

A meta-analysis of tourism related academic journal publications showed that brand personality is one of the most cited personality concepts [17]. Jin-Soo and Back [18] found that competence and sophistication were strongest pillars of upmarket hotel brand personalities. Critics of the brand personality model highlight the aspect that personality is only one part of the overall brand equity. More holistic models are: Brand Asset Valuator [19], BrandZ [20] and Brand Resonance mode [21].

Brand Asset Valuator (BAV) compares the brand equity of thousands of brands across hundreds of different categories. There are four key components of brand equity, according to BAV [19]: Firstly, Energized Differentiation measures the degree to which a brand is seen as different from others, and its perceived momentum and leadership. Secondly, Relevance measures the appropriateness and breadth of a brand's appeal. Thirdly, Esteem measures perceptions of quality and loyalty, or how well the brand is regarded and respected. Fourthly, Knowledge measures how aware and familiar consumers are with the brand.

At the heart of BrandZ model of brand strength is the Brand Dynamics pyramid [20]. According to this model, brand building follows a series of steps. For any one brand, each person interviewed is assigned to one level of the pyramid depending on their responses to a set of questions. The Brand Dynamics Pyramid shows the number of consumers who have reached each level; the highest level being Bonding.

The Brand Resonance model views brand building as an ascending series of steps, from bottom to top by ensuring customers identify the brand and associate it with a specific product class or need firmly establishing the brand meaning in customers' minds by strategically linking a host of tangible and intangible brand associations [21].

Above models are variants of well-known hierarchy of effects models. Aaker's approach [2] is nested within these—it specifies a way brands can establish relevance in the eyes of consumers (via establishing a human identity or character).

Carl Gustav Jung's theory [22] escribed archetypes as the psychic counterpart to physical instincts. Archetypes can be viewed as components of the "collective unconscious, deeply embedded personality patterns that resonate within us and serve to organize and give direction to human thought and action." [22] (p. 77). Initially, CG Jung was a supporter of Freud's theory of the unconscious but later distanced himself from it; the probably most significant difference between Jung and Freud was Jung's concept of archetypes. Richards [23] has traced back the archetype concept from Kant's 'intellectus archetypus' (the purposeful design of all living beings) to Goethe's notion of the 'Urbild' (the original plan of all vertebrate animals).

Jung's work has contributed to contemporary psychology at least one significant aspect: distinguishing between the two major orientations of personality—extroversion and introversion—which is one dimension of the so-called Big 5 [24]. The Myers–Briggs Type Indicator is the best-known personality test and based on Jung's work.

A content analysis of promotions on TV and print media revealed that many brands use archetypal Hero images like the iconic Marlboro Man or Arnold Schwarzenegger's role in Terminator or the figure of Bruce Wayne as Batman [25]. Similar attention got David Beckham's appearance in ads for Adidas, which can be understood to represent viewer's interpretation and unconscious assignment of archetypes. The work of Aaker [2] may also be interpreted to represent images of Freedom, Social, Order and Ego (see Figure 1). There clearly is a psychological component to the effectiveness an ad may have—although, in some cases the appeal of the media selected and the surprising creative are other major variables in terms of attention and engagement.

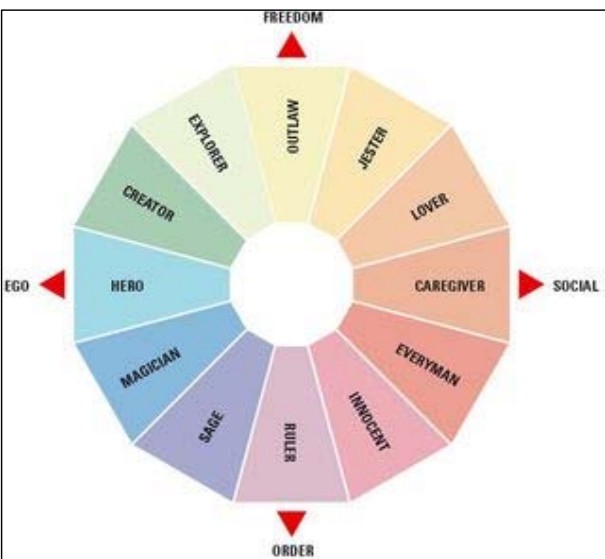

**Figure 1.** Archetypes [26].

Cinderella is another frequently cited example featuring several archetypes [27]. By giving human thoughts and action a direction, archetypes may be building blocks of a successful brand. If this holds true then the use of archetypes can connect deeper and quicker with the psyche of consumers and ultimately lead to purchases of a certain brand [28]. Figure 1 lists the 12 archetypes [22].

Veen [29] and Mark and Pearson [30] showed that the archetype Hero is often used in advertising e.g., for cigarettes and cars. Tsai [31] analysed Nike's Air Jordan in the context of archetypal marketing. He found that the positioning as hero give it a "universal symbolism that all humans may be able to identify" [31] (p. 649). Walle [32] suggested the general use of heroes as archetypes in advertising campaigns. Roberts [33] found different leading archetypes depending on the product category: sports drinks (hero), sports (hero), automobiles (explorer), athletic shoes (explorer), video game consoles (jester), beauty products (lover), soft drink (everyman), beer (everyman) insurance (caregiver), energy drinks (outlaw), apparel (ruler), and political parties (ruler). Faber and Mayer [34] linked different archetypes to individual personalities and their consumer behavior.

Lloyd and Woodside [35] recommend the integration of animals as symbols to activate and connect archetypal associations automatically in consumers' minds, thereby enabling them to activate the cultural schema that the brand represents.

Jung [22] insisted that archetypes stem from a biological and not cultural background. An archetype works in a human being in a similar way as an instinct, as, for example, birds build their nests. However, recent research has shown that archetypes are transmitted more by culture than biology *i.e.*, they are culture specific [36]. To put archetypes into perspective, one has to look at the whole cultural complexity [37]. The symbol of an apple may trigger different associations depending on whether one is a Christian or a Buddhist. Some brands may even change their archetype/brand personality over time.

Cultural differences have been well researched [38–42]. However, not much research has been done on archtypes in an intercultural context. One of the few studies comparing Western and Asian (Indian) perceptions was carried out by Siraj and Kumari [43]. The findings contradict Jung's [22] notion that archetypes are universal. In contrast, Richter *et al.* [44] analyzed individual-level data from 10 countries and identified six common archetypes that are present in all these countries.

Using archetypes in advertising has affinities to mythology, literature and communications. An alternative approach to studying the archetypal aspects of brand image is the literary or cultural view of archetypes, such as the one advanced by Northrop Frye [45], whereby archetypes are seen as a symbol, usually an image, which reoccurs as a pattern to be recognizable as an element of one's literary experience.

Work on narrative theory and characterization in advertising also aligns with the archetype approach [7,9]. Literature text-based analysis can be in form of a semiotic approach (structure seen as inherent in the text) or formalist method (text in the context of images, metaphors, irony, personae *etc.* [46]. The formalist approach has been further developed in the form of a reader-response method within literary criticism, which shows how a text works with the probable knowledge, expectations, or motives of the reader [47].

Stern [46] sees the roots of advertising in medieval allegory. It is very difficult to distinguish between allegory and symbol [48]. For example, the archetype Caregiver could be seen as the symbol or allegory of mother, neighbor, or service provider such as banks or insurances. Most copy platforms of insurances are based on mild fear. The corresponding brand personality dimension is Sincerity. Aghazadeh *et al.* [49] analysed 267 insurance policy holders and found that sincerity affects perceived value and brand loyalty positively.

Similarly, the archetype Hero has been used frequently in medieval allegory [50] with personality dimensions of excitement, sincerity and ruggedness. Personifications such as Lancelot and King Arthur and the Holy Grail come to mind.

## 2. Research Objectives, Framework and Methodology

Our research questions were:

- Is it possible to link Jung's archetypes and Aaker's brand personality framework?
- Are advertising audiences in a position to recognise archetypes?

The objectives of this research were:

(a) to link Aaker's brand personality traits to Jung's archetypes
(b) to analyse personality traits and dimensions that people associate with archetypes
(c) to test these associations on two TV commercials. See Figure 2.

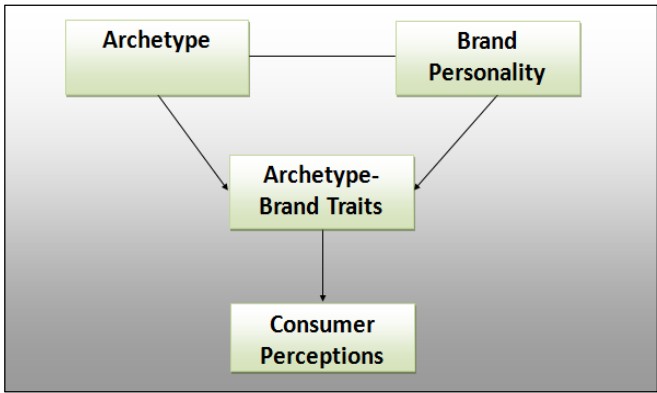

**Figure 2.** Research framework.

To test perceptions on two TV commercials, both using archetypes, an experiment with 47 European and 55 Asian EMBA students was carried out. The facilitator asked to watch two different TV commercials, one from Shangri-la (SL) and one from Banyan Tree (BT) and fill in two surveys. The objective of the first survey was the determination of the archetype used. The second survey looked at brand personality traits of these two TV commercials.

Both, SL and BT are five-star hotel chains in Asia. European students did not know these two hotel chains and therefore were not pre-conditioned in any way. In contrast the 55 Asian students knew the chains which was evaluated by simply asking them in the classroom. The survey was administered using paper and pencil, see Table 2.

**Table 2.** Company demographics (taken from corporate websites).

| Name | Hotels | Employees | Guest Room Nights |
|------|--------|-----------|-------------------|
| SL | 78 | 41,000 | 7.5 Million |
| BL | 31 | 15,000 | 3 Million |

Both commercials are without a single word of dialogue. The SL commercial did not feature any SL product or service and only linked the logo to the message at the very end. BT used another approach. Throughout the whole commercial, products and services of BT were shown and linked to one message: BT stands for charming and welcoming service. The main theme was the hospitality in the form of an upmarket spa and relaxation at a private swimming pool. Beds were decorated with red roses and a harmonic young Asian couple enjoying their romantic time. In contrast, SL featured wolves in its "It's in our nature" campaign. In the TV commercial, a stranger wanders through snow covered mountains and gets lost. The wolves surround the tired traveler and warm him with their body heat. They are the real heroes of the story. The SL ad was slightly out-of-the-box because it did not fit traditional hotel advertising showing facilities and service. Instead it featured wolves that are not generally known to be hospitable and amicable to humans. See Figures 3 and 4.

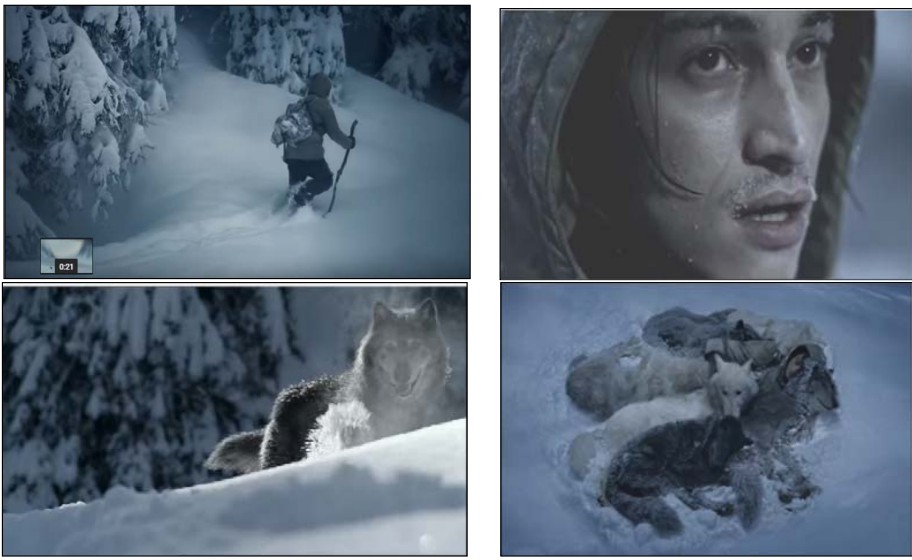

**Figure 3.** Screenshot TV commercial SL.

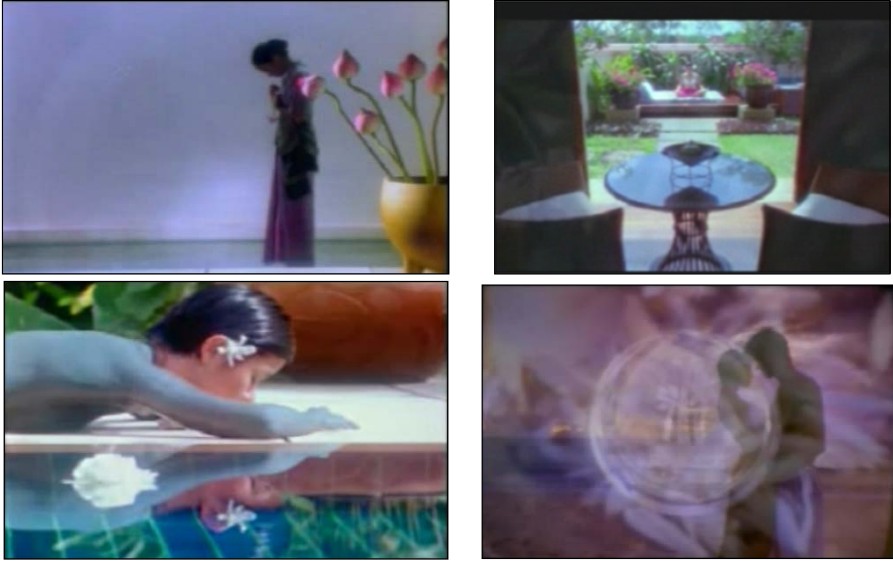

**Figure 4.** Screenshot TV commercial BT.

To evaluate which archetype people saw in the ads a test similar to Characterlab.com's test was designed which in turn is based on the Pearson-Marr Archetype Indicator [51]. After viewing the ads the participants had to rank attributes that described the ads. The survey was administered in the classroom using paper and pencil. Instead of asking directly what kind of archetype the participants saw, viewers had to rate attributes along three dimensions: Look, Feel and Talk. For example, fantasy landscape/creatures stood for Magician, pleasurable sensations for Lover on the Look dimension. On the Feel dimension the attributes enigmatic, transformational, mysterious and amazing stood for Magician whereas passionate, elegant *etc.* stood for Lover. Speaking about sensory experience stood for Lover on the Talk dimension. All in all five attributes per (12) archetype times three dimensions (look, feel, talk) were analyzed. The question the participants were asked on the Look dimension: "Thinking about what you saw in the commercial, please review the ad and give your overall impression of the SL/BL" 12 cards were handed out with each card having some explanatory text, e.g., for Caregiver "Caring staff, warm, comforting environment, loving, embraced, home-made food, and comfort." On the feel dimension the question was: "Thinking about how the BL/SL commercial makes you

feel and the emotions it evokes." With choices e.g., for Caregiver "caring, supportive, protective, compassionate, selfless, comforting, and nurturing." On the Talk dimension: "Thinking about how SL/BL would speak to you if it were a person," e.g., for Caregiver "protect, care, help, safe, look after, and reliable."

The result showed that the predominant archetype for SL was Hero and for BT Lover. As next step we tried to link archetypes to brand personalities by using Aaker's personality traits as attributes of archetypes, see Table 3. The authors picked the top three traits that best fit Jung's [22] description of each of the 12 archetypes. Whereas the first two columns of Table 3 are based on Jung's terminology, the third and fourth column use Aaker's [2] brand personality terminology. The matching between Jung's and Aaker's categories were done by the authors, as such they are subjective.

**Table 3.** Archetypes and brand personality.

| Archetype | Archetype Manifestation | Personality Trait | Brand Personality Dimension |
|---|---|---|---|
| Ruler | Stability | Reliable<br>Tough<br>Upper Class | Competence<br>Ruggedness<br>Sophistication |
| Creator | Stability<br>Independence | Imaginative<br>Unique<br>Upper Class | Excitement<br>Excitement<br>Sophistication |
| Caregiver | Stability<br>Belonging | Embraced<br>Welcoming<br>Genuine | Sincerity<br>Sincerity<br>Sincerity |
| Jester | Belonging<br>Mastery | Genuine<br>Charming<br>Imaginative | Sincerity<br>Sophistication<br>Excitement |
| Lover | Belonging<br>Stability | Welcoming<br>Charming<br>Embraced | Sincerity<br>Sophistication<br>Sincerity |
| Regular Guy | Belonging | Welcoming<br>Reliable<br>Genuine | Sincerity<br>Competence<br>Sincerity |
| Outlaw | Mastery<br>Independence | Adventure<br>Tough<br>Charming | Excitement<br>Ruggedness<br>Sophistication |
| Magician | Mastery<br>Belonging | Embraced<br>Reliable<br>Imaginative | Sincerity<br>Competence<br>Excitement |
| Hero | Mastery | Adventure<br>Genuine<br>Tough | Excitement<br>Sincerity<br>Ruggedness |
| Sage | Independence<br>Stability | Unique<br>Reliable<br>Imaginative | Excitement<br>Competence<br>Excitement |
| Explorer | Independence | Adventure<br>Unique<br>Tough | Excitement<br>Excitement<br>Ruggedness |
| Innocent | Independence<br>Mastery | Genuine<br>Unique<br>Reliable | Sincerity<br>Excitement<br>Competence |

## 3. Findings

Equal weight was given to the three personality traits and averages calculated on a 1–5 Likert scale, see Table 4.

**Table 4.** Perceived personality traits averages.

| Archetype | Personality Traits | | | SL Mean | BL Mean |
|---|---|---|---|---|---|
| Ruler | Reliable | Tough | Upper Class | 3.56 | 3.45 |
| Creator | Imaginative | Unique | Upper Class | 3.77 | 3.52 |
| Caregiver | Embraced | Welcoming | Genuine | 3.79 | 3.60 |
| Jester | Genuine | Charming | Imaginative | 3.85 | 3.48 |
| Lover | Welcoming | Charming | Embraced | 3.77 | 3.81 |
| Regular Guy | Welcoming | Reliable | Genuine | 3.70 | 3.65 |
| Outlaw | Adventure | Tough | Charming | 3.72 | 3.08 |
| Magician | Embraced | Reliable | Imaginative | 3.92 | 3.50 |
| Hero | Adventure | Genuine | Tough | 3.74 | 2.87 |
| Sage | Unique | Reliable | Imaginative | 3.81 | 3.44 |
| Explorer | Adventure | Unique | Tough | 3.70 | 2.90 |
| Innocent | Genuine | Unique | Reliable | 3.70 | 3.43 |

The result confirmed the positioning that the BT commercial stood for Lover. In SL's case this was not the case. However, Hero and Magician are not far apart, see Figure 1. Both share 'Ego' as common driving force, so-called cardinal orientations [22]. One reason for not recognizing the Hero may be that most people would not associate wolves with heroic behavior. On the other hand, looking at the personality traits of Hero (daring/adventure, genuine, tough) it may well go with wolves. To analyze this aspect further we analysed the rated (1–5 Likert scale) personality traits in form of a Factor Analysis (Principal Component with Varimax Rotation), see Table 5.

**Table 5.** SL Personality traits.

| Traits | Component (Rotated) | | | |
|---|---|---|---|---|
| | Hero | Hero | 3 | 4 |
| SL Genuine | 0.859 | 0.192 | −0.031 | 0.223 |
| SL Adventure | 0.823 | −0.166 | 0.058 | 0.038 |
| SL Tough | 0.268 | 0.809 | −0.049 | −0.014 |
| SL Reliable | 0.437 | 0.632 | −0.069 | 0.378 |
| SL Welcoming | 0.478 | 0.623 | 0.309 | −0.014 |
| SL Imaginative | −0.057 | −0.113 | 0.851 | 0.162 |
| SL Charming | 0.036 | 0.369 | 0.628 | 0.365 |
| SL Embraced | 0.374 | 0.414 | 0.603 | −0.284 |
| SL Unique | 0.243 | 0.012 | -0.004 | 0.821 |
| SL Upper Class | −0.035 | 0.098 | 0.396 | 0.708 |

Factor 1, explaining 32.13% of variance, showed high loadings of genuine and adventure. Factor 2, explaining 16.49% of variance, had tough as high loading trait. Taking these three together they constitute the personality traits of Hero. In essence, the perceived archetype of the SL commercial, using an Archetype Indicator, is Hero but when using Aaker's personality traits [2] it comes up as Magician which is not too different from Hero. In the factor analysis, the three Hero traits explained most of the variance.

## 4. Implications

Using the Brand Personality approach we were able to link personality traits to archetypes and can confirm the positioning in one case. Archetype positioning was measured by using three dimensions (look, feel, talk) with several attributes describing each archetype. To arrive at reliable results the cultural factors have to be introduced into the equation. The charming Thai lady featured in the BT commercial is seen as rugged by some Asian viewers but utterly charming by almost all Europeans.

Since advertising should use Hero as an archetype [30] the intended positioning may not arrive at the consumer level because of ambiguous symbols such as the wolf pack in the SL commercial.

Aaker's [2] brand personality model can help to clarify the traits of Hero brands. The horse used in the iconic Marlboro ads may have been a more suitable animal for a Hero than wolves.

Veen [29] demonstrated that Hero can be a powerful archetype in certain product categories such as cigarettes and cars. Whether Hero is the preferable archetype in the hospitality sector remains to be seen. BT's Lover positioning may be more appealing to hotel guests despite its very common theme. One could imagine that Outlaw, breaking the rules, is another suitable archetype candidate for hotel advertising besides Lover and Hero. Because of the confidentiality of data we could not measure and compare the effectiveness of both campaigns.

Our aim was to revitalize the concept of archetypes by combining it with the more contemporary concept of brand personality. The implication for advertisers is that Jung's archetypes should be an essential part of an advertising agency briefing. Game changer sites such as Airbnb may want to be seen as Outlaws and established hotels maybe Lovers. The polarization that Outlaw is high on the freedom dimension and Lover on the social dimension in Jung's framework is somewhat too vague for an advertising agency. Attributing archetypes with personality traits represents a more hands-on approach. A genuine, adventurous, tough brand personality is more specific than just defining Hero as brand personality. On the other hand, a simple character may summarize a brand more concise than many words e.g., the Marlboro man as Hero. Archetypes can come in blended form *i.e.*, being a Hero does not exclude being an Outlaw at the same time [30].

Our research has shown that both frameworks can be combined. Further research should look into deeper psychological and cultural understanding of archetypes and move beyond brand personality traits by linking it to a host of tangible and intangible intercultural brand traits and associations. Measuring archetypes by neurophysiological methods to better understand the impact of promotion on attention, affect, memory, and desirability could be another interesting research area. Advertising effectiveness could be measured by: self-reporting, implicit measures, eye tracking, biometrics, electroencephalography, and functional magnetic resonance imaging.

Instead of asking participants what personality traits they saw, one could think of a content analysis of the commercials. With the help of open source systems such as Solr and Lucene the actual content can be analyzed. However, it will still need a human to classify terms. For example: "Jung influenced Page." Does Jung stand for CG Jung and Page for Google's Larry Page or Jimmy Page, guitarist from Led Zeppelin? Does Jung stand for Sungha Jung, a Korean musician? So far, only humans can assess the meaning when looking at the context. The times when a system can analyze a TV commercial and extract brand personality traits and archetypes automatically are still a few years off. There are limitations to our research such as the small sample size. We consider our findings as preliminary and encourage other researchers to conduct own experiments. Further research should look into the impact of ad campaigns (sets of related ads) in delivering a consistent (reliable) impression.

## 5. Conclusions

Although 43 years separate Jung's and Aaker's framework we feel that by aligning archetypes and brand personalities the body of management knowledge will be broadened by helping advertisers to define their campaign objectives in another deeper dimension.

**Author Contributions:** All authors contributed equally to this article. All authors collected and analysed data and contributed to preparing the manuscript.

**Conflicts of Interest:** The authors declare no conflict of interest.

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
