# Peer review of "Advertising between Archetype and Brand Personality"

_admsci, doi:10.3390/admsci6020005_

Round 1

Reviewer 1 Report

Overall, I like the basic idea of this paper: archetypes and brand personality can be used together to create distinct brand images that differentiate and position offerings in a marketplace. These ideas in developed with (some of) the relevant literature and are demonstrated using a simple two-firm case study. That being said, I believe there is an opportunity to improve the paper by developing the conceptual foundations more thoroughly and my being more transparent and clear in the methodology section. To that end, I offer the following constructive suggestions:

Conceptual Foundations

(a)    Why Jung? The authors have selected Jung’s archetypes as the theoretical foundation for the paper. While Jungian approaches have been used by marketing and consumer researchers in the past (e.g., Hirschman), the reasons for this choice need to be justified in the context of other alternatives. Jungian approaches suffer from two core limitations: (a) the assertion of universality – which the authors note, and (b) the psychoanalytic heritage of the ideas (which presupposes a certain view of motivation) is disavowed by most contemporary psychologists.

(b)   Literary archetypes: An alternative approach to studying the archetypal aspects of brand image can be found by scholars who adopt a literary or cultural view of archetypes, such as the ones advanced by Joseph Campbell or Northrop Frye (see, for example: Stern 1989). One could make the argument that using archetypes in advertising has greater affinities to mythology, literature and communications. Work on narrative theory and characterization in advertising also aligns with the archetype approach (see, for example: Mulvey and Medina 2003; Padgett and Allen 1997; Scott 1994; Stern 1988). This paper’s potential impact will be enhanced if it can find a way to align itself with this growing branch of advertising research.

(c)    The abstract states: “Aaker is seen by many as the branding guru” and refers to Aaker (1997) in the introductory paragraph. Are you referring to David Aaker (the father) or Jennifer Aaker (the daughter)?

(d)   Indeed, Aaker (1997) is a highly-cited work. Are there any other limitations or criticisms the reader should be informed of? There has been some backlash to the application of brand personality to inanimate objects. This may be of concern, because the paper uses hotels as an example. Personally, I agree with you – brand personality is a key facet of brand identity. However, you may want to tell readers how brands can convey aspects of personality via advertising (see: Allen and Olson 1995). This is highly relevant to answering the “so what?” question that is addressed in the implications section.

(e)   Page 2, line 62: I disagree with the claim that “above models can be seen as extensions of Aaker’s (1997) brand personality model.” The models you listed are variants of well-known hierarchy of effects models. Aaker’s approach is nested within these – it specifies a way brands can establish relevance in the eyes of consumers (via establishing a human identity or character).

Methodology

(a)    Though the paper is exploratory, I am not convinced that it aligns with the “grounded theory” tradition. Grounded theory is based on the idea of immersion in data to induct new conceptual categories. I see the present study as using existing typologies and categories – there are no new concepts that emerge from the research that can be applied directly to future research.

(b)   Sample size: A critical view would state that the present research relies on a very small sample of advertisements (n=2) to make its claims. After all, the research focuses on the capacity of an ad to convey archetypal and personality-based aspects of brand identity. The results would have been more compelling if the study measured the impact of ad campaigns (sets of related ads) in delivering a consistent (reliable) impression. Though such studies are rare, good interpretive and quantitative examples can be found in the literature. At a minimum, this exploratory paper should set a more specific and ambitious plan for systematic future research on the topic. For example, Padgett and Mulvey (2007) illustrate a method to identify personal values conveyed by 16 ad campaigns (another complementary dimension of personified brands) and Padgett and Mulvey (2009) demonstrate a novel way to characterize customer-brand relationship archetypes. Imagine a conversation with these authors: what do you contribute to the ongoing conversation?

(c)    Evidence of claim? Page 4, line 124 states: “European students did not know these two hotel chains and therefore were not pre-conditioned in any way. In contrast the 55 Asian students knew the chains.” Was brand familiarity actually measured, or was it assumed? The claim is very absolute – I wonder if any of them ever travelled internationally – presumably EMBAs would be more worldly or cosmopolitan than this…

(d)   Figure 3: Because you are studying commercials (which unfold over time), I don’t think the single screenshots really add much value to the paper. However, if you included a storyboard (set of screenshots) for each ad, that would help the reader (for examples, see: Mulvey and Medina 2003; Scott 1994).

(e)   Clarity and elaboration of Page 5+: The methods section lacks clarity – it would be very difficult for a researcher to replicate the procedure. Please provide greater detail of the scales used, the anchor terms, and how the survey/rating task was administered (online? Paper and pencil?).

(f)     “Archetype Articulate Charisma” (heading title in Table 3): This is not clear. Either define the term in the text or use phrasing that is clear to the reader. Please don’t expect them to consult original sources by Jung!

(g)    Findings: Who rated the three personality traits? The student/respondents? How many items were used? Why do you not report reliability measures for the items and scale?

(h)   Factor analysis: Have you reported principal components, or a rotated solution? More details are required.

(i)      Interpretation: The interpretation of Archetypes seems to assume that the types are mutually-exclusive. However, this is not really the case. A close reading of some of the cited work (i.e., Mark and Pearson 1991) recognizes that archetypes can be blended.

As mentioned at the onset of my review, I fully support the authors’ aim to revitalize and blend archetypal and personality-based approaches to brand identity. I hope my suggestions are given full consideration, as I believe that they will assist them in achieving this important objective.

REFERENCES

Allen, Douglas E and Jerry Olson (1995), "Conceptualizing and Creating Brand Personality: A Narrative Theory Approach," Advances in Consumer Research, 22 (1), 392-93.

 Mulvey, Michael S. and Carmen Medina (2003), "Invoking the Rhetorical Power of Character to Create Identifications," in Persuasive Imagery: A Consumer Response Perspective, ed. Linda M. Scott and Rajeev Batra, Mahwah, NJ: Lawrence Erlbaum Associates, 223-45.

Padgett, Dan and Douglas Allen (1997), "Communicating Experiences: A Narrative Approach to Creating Service Brand Image," Journal of Advertising, 26 (4), 49-62.

 Padgett, Dan and Michael S Mulvey (2009), "Experiential Positioning: Strategic Differentiation of Customer-Brand Relationships," Innovative Marketing, 5 (3), 87-95.

Padgett, Dan and Michael S. Mulvey (2007), "Differentiation Via Technology: Strategic Positioning of Services Following the Introduction of Disruptive Technology," Journal of Retailing, 83 (4), 375-91.

Scott, Linda M. (1994), "The Bridge from Text to Mind: Adapting Reader-Response Theory to Consumer Research," Journal of Consumer Research, 21 (3), 461-80.

Stern, Barbara B. (1988), "Medieval Allegory: Roots of Advertising Strategy for the Mass Market," Journal of Marketing, 52 (3), 84-94.

 --- (1989), "Literary Criticism and Consumer Research: Overview and Illustrative Analysis," Journal of Consumer Research, 16 (3), 322-34.

Reviewer 2 Report

I think there are some ways in which you may tighten up the description of the experimental design.The comparisons between Jung and Aaker are always interesting - although the small sample size may be troubling to some.  Reviewing the famous Marlbro ads or thinking about the attention David Beckham's appearance in ads for Adidas receives can be understood to represent viewer's interpretation and unconscious assignment of famous archetypes. It is interesting and the work of Aaker may also be interpreted to represent images of Freedom, Social, Order and Ego.  There clearly is a psychological component to the effectiveness an ad may have - although in some cases the appeal of the media selected and the surprising creative are major variables in terms of attention and engagement.

I think that the relevance of the research to practitioners as well as academics needs to include some current findings about ads, their placement and effectiveness. For example the fact that some "heroes" can move across categories - e.g., Tiger Woods for Buick. Time of message delivery and personal relevance to the targeted audience may be ( I believe) of more importance than the creative.  The general and growing aversion to ads - ad blocker or the increased use of native advertising  - sponsored branded content - that appears in desired media channels is a major issue for ad agencies.  I hope the authors find these comments helpful.

Round 2

Reviewer 1 Report

The authors have done a commendable job addressing my comments and concerns. I am particularly pleased with the effort made to improve the conceptual parts of the paper. The methods and reporting of results are improved. Personally, I would have preferred to see more detail - but this concern may reflect my taste rather than a flaw or shortcoming of the paper. Good work! I hope other readers enjoy reading your ideas as much as I did.